# Emissive brightening in molecular graphene nanoribbons by twilight states

**Bernd K. Sturdza** [1] ✉, **Fanmiao Kong**[2], **Xuelin Yao**[2], **Wenhui Niu**[3,4], **Ji Ma** [3,4], **Xinliang Feng**[3,4], **Moritz K. Riede** [1], **Lapo Bogani** [2,5] ✉ **& Robin J. Nicholas** [1] ✉

Carbon nanomaterials are expected to be bright and efficient emitters, but structural disorder, intermolecular interactions and the intrinsic presence of dark states suppress their photoluminescence. Here, we study synthetically-made graphene nanoribbons with atomically precise edges and which are designed to suppress intermolecular interactions to demonstrate strong photoluminescence in both solutions and thin films. The resulting high spectral resolution reveals strong vibron-electron coupling from the radial-breathing-like mode of the ribbons. In addition, their cove-edge structure produces inter-valley mixing, which brightens conventionally-dark states to generate hitherto-unrecognised twilight states as predicted by theory. The coupling of these states to the nanoribbon phonon modes affects absorption and emission differently, suggesting a complex interaction with both Herzberg–Teller and Franck– Condon coupling present. Detailed understanding of the fundamental electronic processes governing the optical response will help the tailored chemical design of nanocarbon optical devices, via gap tuning and side-chain functionalisation.

Most semiconducting carbon nano-structures have direct band gaps and extreme quantum confinement, so they would be expected to yield superb photoluminescence (PL) quantum efficiencies (PLQEs) from their excitons which have large binding energies[1–3]. Alas, non-radiative dark states, disorder, and interactions among the carbon nano-structures all lead to substantial PL quenching[4–7], limiting their potential for applications in bio-imaging, solar energy conversion, in opto-electronics and in display devices, as well as in quantum technologies for single-photon sources, up-conversion, and sensing[2,3,8,9]. The most appealing systems, those with large exciton-diffusion coherent lengths, such as carbon nanotubes (CNTs), are actually those most affected. CNTs need de-bundling, but covalent functionalisation breaks the π-conjugation, and non-covalent surfactants are randomly arranged around the CNT surface, so both mechanisms introduce disorder and severely disrupt electronic properties, limiting the PLQE[7,10,11].

Moreover, the presence of degenerate valley states introduces a fundamental limitation by producing dark excitons in CNTs and graphene. Dark excitons occur in carbon nanomaterials because momentum conservation allows PL only from states that have zero net angular momentum. In semiconducting CNTs, the multiple K-points of the graphene Brillouin zone lead to the formation of a dark valley doublet state and two coupled singlets, only one of which is bright (Fig. 1a). Graphene nanoribbons (GNRs) with zigzag edges are largely equivalent to armchair CNTs: their edges generate a bandgap with the typical bright and dark states[12,13].

Here we show, when the edge structure of the molecular GNR is engineered so as to introduce a periodic modulation of the edge states as in cove-edged GNRs (Fig. 1a), important changes occur to the optical properties. We demonstrate that this causes brightening of the photoluminescence and show how it is linked to electron-phonon coupling.

[1]Clarendon Laboratory, Department of Physics, University of Oxford, Parks Road, Oxford OX1 3PU, United Kingdom. [2]Department of Materials, University of Oxford, 16 Parks Road, Oxford OX1 3PH, United Kingdom. [3]Center for Advancing Electronics Dresden (CFAED), Faculty of Chemistry and Food Chemistry, Technische Universität Dresden, Mommsenstrasse 4, 01062 Dresden, Germany. [4]Max Planck Institute of Microstructure Physics, Weinberg 2, 06120 Halle, Germany. [5]Departments of Chemistry and Physics, University of Florence, V. della Lastruccia, 50019 Sesto Fiorentino, Italy. ✉e-mail: bernd.sturdza@physics.ox.ac.uk; lapo.bogani@materials.ox.ac.uk; robin.nicholas@physics.ox.ac.uk

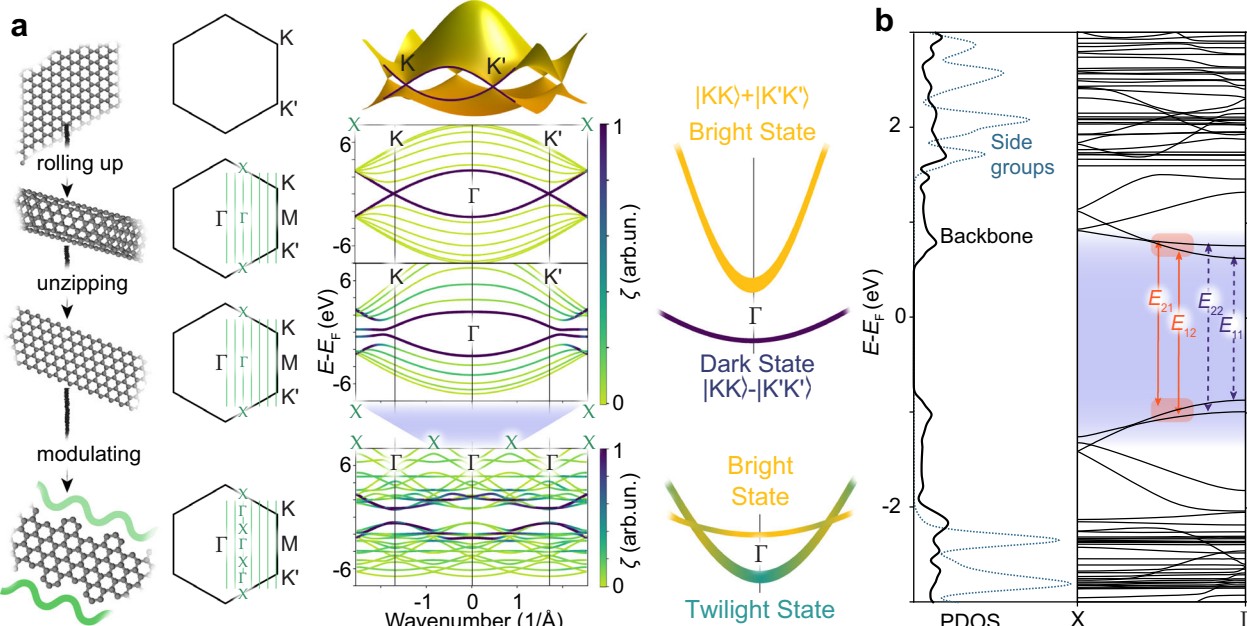

**Fig. 1 | Bright, Dark and Twilight States. a** Comparison of graphene, a (6,6) carbon nanotube, a zigzag GNR and a cove-edge GNR, showing the real and reciprocal lattices, with $K$, $K'$, $M$, $\Gamma$ and $X$ points indicated, together with the discretization produced by lateral confinement (green lines). The bands of each nanostructure are coloured based on their projection $\zeta$ of each point onto the graphene state along the rightmost green k line that passes through $K - M - K'$, resulting in bright and dark states, from the linear combinations of valleys, or yielding the brighter twilight states for cove-edge GNRs. **b** Zoom-in of the projected density of states (PDOS) and band structure of a synthetically-achievable cove-edge GNR, including the contributions from the backbone and the side groups as per Fig. 2a. Transitions are labelled by their main inter-band contribution.

## Results and discussion

### Edge engineering and twilight states in cove-GNR

With a threefold edge symmetry, zone-folding of the dark states into the $\Gamma$-point zone centre allows all of the previously-dark excitons to mix and become much more emissive (Fig. 1a), increasing the theoretical limit of the PLQE fourfold. To distinguish them from conventional bright states, we dub these new long-lived emissive states that arise from previously-dark ones as twilight states. The projection of the wave-function of several different carbon nanostructures $\Psi_{n,k}$ onto the graphene wave-function $\Psi_{n,k}^{gra}$, called $\zeta = \| \langle \Psi_{n,k}^{gra} | \Psi_{n,k} \rangle \|$ offers a quantitative insight into the nature of these new states (Supplementary Text 2). A plot of $\zeta$ for each wave-vector $k$ and band $n$ shows that armchair CNT and zigzag GNR states project almost perfectly onto the original graphene ones, and thus form the usual bright and dark excitons, while valleys become very strongly mixed in cove-edged GNRs (Fig. 1a). Now, not all states can be projected onto combinations of graphene ones, and $k = 0$ relaxation to the ground state by photon emission is always allowed (Fig. 1a). The projected density of states for these chemically-feasible GNRs shows a low-energy spectrum dominated by the GNR backbone (Fig. 1b). The absorption no longer belongs to well-defined interband transitions, but to groups of closely-lying transitions near the $\Gamma$ point[14]. In particular, the energies of $c_1 \leftarrow v_2$, $c_2 \leftarrow v_1$ electronic transitions are close, and can only be labelled by their dominant contribution[14].

### Enhanced luminescence and solubility from large side groups

Even so, GNRs remain plagued by the other issues that affect CNTs, and, to date, spectra of GNRs in solution have remained broad and featureless[15], an indication of inter-ribbon exciton transfer or edge and width disorder[16]. Finer spectral features are available only in conditions incompatible with applications, such as single ribbons produced by surface synthesis under high-vacuum[13].

We overcome these issues by using cove-edged GNRs functionalised with bulky groups, here cycloadducts from the Diels–Alder reaction of anthracene with $N$-$n$-octadecylmaleimide. The GNRs still show a length dispersion, but this has little effect on the optical response, as the width-length aspect ratio is large[17] and contrary to carbon nanotubes, all ribbons have the same width. We focus on suppressing the electronic interactions among ribbons arising from aggregation. The two possible solubilising groups are covalently attached to the edges with a grafting ratio of 77%[17] and spaced entirely regularly at the GNR edges (Fig. 2a), thus minimizing disorder[18].

The radius of the bulky groups is approximately 0.5 nm, much larger than the interlayer spacing of graphite, and so they suppress $\pi$-$\pi$-stacking interactions among GNRs which are thus debundled. These side groups may still lead to some weak ribbon-ribbon interactions, but the suppression of the stacking interaction effectively isolates the electronic states of the ribbons, creating a dramatic effect on the optical response (Fig. 2b). The resulting samples are remarkable: they display exceptional solubility in organic solvents, such as toluene, tetrahydrofuran, and chloroform, remain stable for at least 16 months without precipitation, and thin film samples with identical optical behaviour can be produced by inclusion inside transparent polymer matrices, see materials and methods. The improvements in the optical response are striking: the PL emission is exceptionally bright, with well-defined features (Fig. 2d). By contrast, in non-debundled samples of cove-edged GNRs functionalised with the usual dodecyl groups (Supplementary Fig. 1), which are much less effective at separating the GNRs, we only see broad and featureless photoluminescence (PL) peaks, similar to previous literature[16] (Supplementary Fig. 2) with around 3 orders of magnitude lower peak intensities (Fig. 2b).

### Phonon modes revealed by Raman scattering

Raman scattering reveals the phonon modes of the new debundled cove-edged GNRs. The radial-breathing-like-mode (RBLM) (31.6 meV) becomes the dominant Raman feature, with a weaker overtone at twice the energy and two weaker peaks at 15 and 20 meV, likely associated with out-of-plane and longitudinal acoustic modes[19,20], also visible

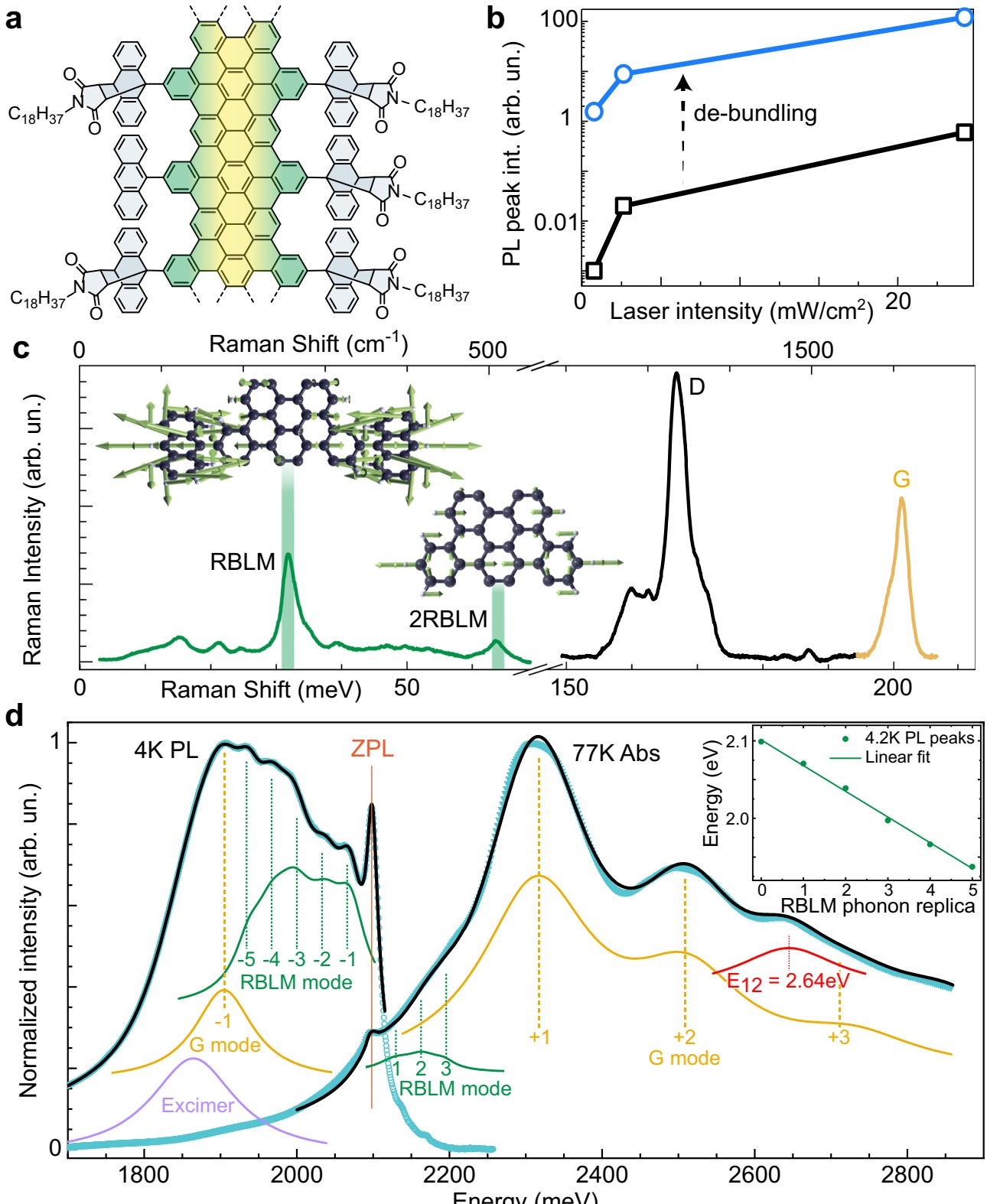

**Fig. 2 | Photoluminescence enhancement and spectroscopic features.**
**a** Structure of molecular GNRs. The GNR backbone is highlighted in yellow-green, and the de-bundling side groups in grey. **b** Comparison of the photoluminescence peak intensities for bundled and debundled samples, acquired under the same conditions. **c** Raman spectrum, acquired on thin films at 2.33 eV excitation. The G, D, radial-breathing-like mode (RBLM) and its overtone are labelled, together with calculated atomic displacements, including anthracene side groups for the RBLM,

and only the backbone for the overtone. **d** Photoluminescence and absorption spectra of GNR thin films at 4 K and 77 K (dots) and modelling (black lines). Individual contributions are shown for RBLM (green) and G (yellow) modes, as well as the $E_{12}$ interband transition (red) and excimer emission (violet). The inset shows the fitted peak positions of the narrow PL features with a linear fit returning the RBLM energy (32.5 meV).

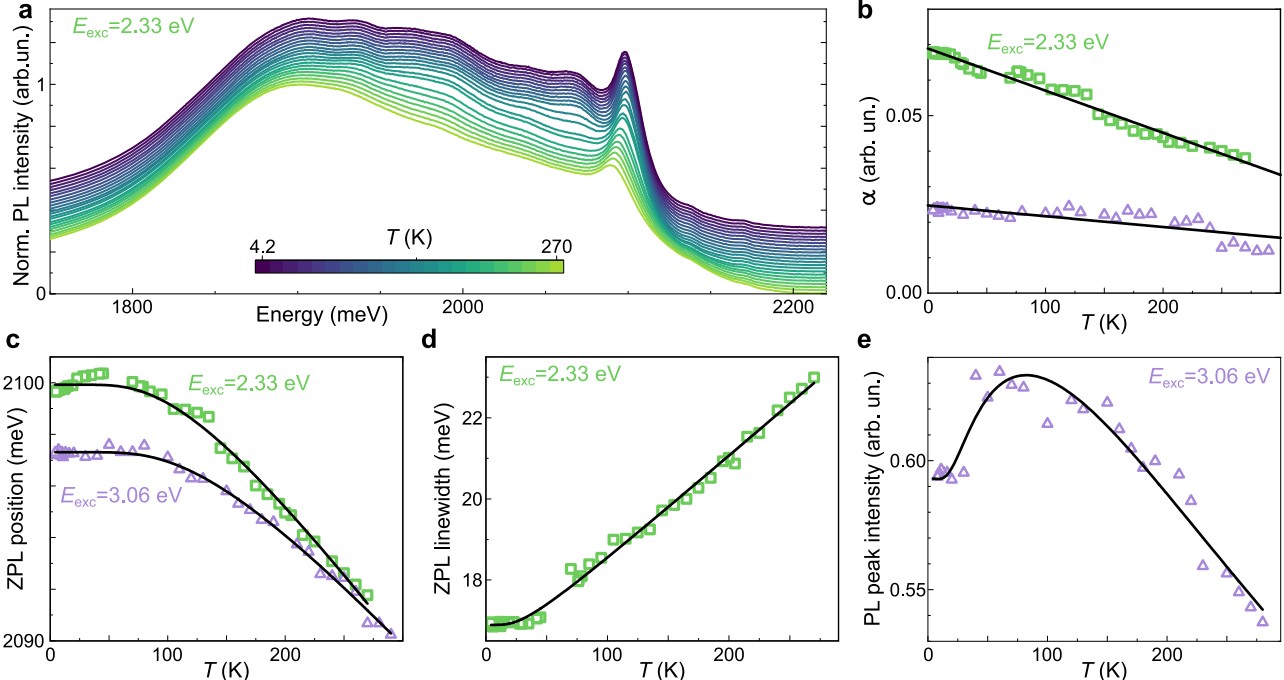

**Fig. 3 | Temperature dependence. a** PL spectra acquired on thin films at different $T$ (colour bar), with photoexcitation at 2.33 eV. **b** Debye–Waller factor $\alpha$ as a function of $T$ for photoexcitation at 2.33 and 3.06 eV (green and violet, respectively). Lines are linear regressions. **c** Zero-phonon-line position vs $T$ for photoexcitation at 2.33

and 3.06 eV (green and violet). Lines are fitted with the Fan equation. **d** Temperature evolution of the zero-phonon peak FWHM under 2.33 eV excitation shown with line fit (see text). **e** $T$-dependence of the PL peak intensity under 3.06 eV excitation, shown with a two 1D exciton fit (see text).

(Fig. 2c). A lateral zone-folding model, insensitive to the edge structure[20,21], yields a RBLM energy $E_{RBLM} = \hbar\omega_R = \frac{399.5 \text{Å meV}}{w_{GNR}} = 32$ meV, which matches observations. Local density approximation calculations provide the vibrational symmetries and reproduce quantitatively the vibrational response, with the calculated RBLM energy, 32.2 meV, matching the main Raman peak (Fig. 2c). It is important to note that the symmetry of the RBLM produces a modulation of the GNR width, with large displacements at the edges, and is thus likely to couple strongly to electronic modes. The higher energy D (166 meV) and G (200 meV) Raman modes[22] are as expected, but their higher order modes are hidden by the much brighter PL and only emerge at low-$T$ due to the narrowing PL emission. By contrast, the higher-order D and G modes can be easily seen in the bundled GNRs above the very weak PL (Supplementary Fig. 3).

**Low temperature PL and electron-phonon coupling**

The high resolution of the optical response obtained allows a complete unravelling of the mechanisms at play and indeed reveals the fundamental importance of the interplay of electronic and vibrational states. The absorption spectrum of the GNRs consists of three wide peaks at 2.29 eV, 2.49 eV and 2.63 eV plus a narrower feature at 2.09 eV (Fig. 2d). The overall PLQE for the de-bundled GNRs reaches a value of 6.4 ± 0.2% at room temperature (Table S1), with a gradual enhancement of the emission on lowering the temperature $T$ (Fig. 3e), so that the total PL intensity increases by 20%, leading to a remarkably high overall 8 ± 0.5% PLQE already beyond the theoretical limit of 6% for CNTs and conventional GNRs[12].

The 2.09 eV feature is present in both absorption and emission at the same energy, becomes more visible upon lowering $T$ (Fig. 3a, plotted vs wavelength in Supplementary Fig. 4), and is thus the zero-phonon line (ZPL) of the inter-band transition. This observation is exceptional for ensemble measurements on carbon nanostructures, where the ZPL is usually broadened by polydispersity, disorder in the individual nanostructures such as edge and width fluctuations, and inter-structure energy transfers. The optical response therefore

demonstrates the superb level of cleanliness of the de-bundled GNRs. We can thus directly probe the $T$-evolution of the semiconducting bandgap ($\Delta_G$) from the ZPL position (PL for 3.06 eV excitation, Supplementary Fig. 5). The low-$T$ non-monotonic behaviour (Fig. 3c) resembles that observed in semiconducting CNTs[23], with the slight decrease in the 2.33 eV data probably due to increasing emission from the twilight states at slightly lower energy. It is important to notice that, as in other carbon-based nanomaterials[23], the total shift is around 10 meV, much larger than the 1 meV effect produced by thermal expansion. The behaviour shows good agreement with what is expected for semiconductors, where[24] $\Delta_G(T) = \Delta_G(0) + A\left(1 - e^{E_{ph}/k_B T}\right)^{-1}$, where $k_B$ is the Boltzmann constant, $A$ is the Fan parameter, and the Bose-Einstein statistical factor contains the characteristic phonon energy $E_{ph}$[25]. We obtain $E_{ph}$ = 30 ± 2 meV and 35 ± 2 meV for the two excitation wavelengths, close to the RBLM energy, revealing that phonon coupling to the RBLM is the key mechanism modulating $\Delta_G$, as also backed up by the $T$-dependence of the D and G Raman lines (Supplementary Fig. 6). Intuitively, this is expected: the RBLM is the one altering the GNR width, responsible for the electronic confinement.

The $T$-dependence of the line-broadening parameter $\Gamma(T)$ of the ZPL reveals what phonons play a major role. Fitting with the expression $\Gamma(T) = \Gamma_0 + \Gamma_{ph} = \Gamma_0 + \gamma\left(e^{E_{ph}/k_B T} - 1\right)^{-1}$, yields the inhomogeneous broadening linked to scattering on terminations[26], $\Gamma_0 = 16.89 ± 0.03$ meV, the homogeneous broadening produced by phonon scattering[27] $\Gamma_{ph}$, and an electron-phonon coupling strength $\gamma = 2.1 ± 0.3$ meV with phonons at energy $E_{ph} = 7.1 ± 0.7$ meV. Interestingly, these are the very same transverse bending modes that strongly modulate the electron transport in transistors made from the same GNRs[17].

A similar energy is obtained for the gap between bright and twilight states, $\Delta$, as extracted from the ZPL peak intensity vs $T$ (Fig. 3e). Assuming exciton thermalisation, the PL intensity $I$ can be described as[5] $I \propto \frac{1}{(T^2 + T_0^2)^{1/4}} \cdot \frac{m + e^{-\Delta/kT}}{1 + e^{-\Delta/kT}}$ where the power-law dependence is characteristic of one-dimensional excitons, $m$ describes the spectral weight of the excitonic states[4] and $T_0$ is a $T$-independent Lorentzian

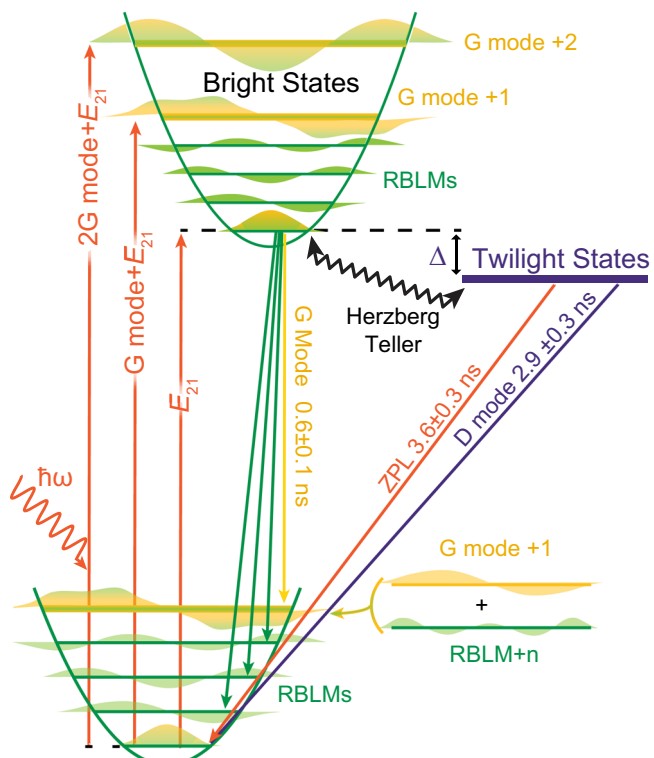

**Fig. 4 | Energy diagram for cove-GNR.** Cove-GNR transitions, with G modes in yellow, RBLMs in green (G and RBLM mixing is schematically depicted) and the twilight states in violet.

broadening term. The intensity maximum around 75K indicates the presence of two sets of excitonic states separated by $\Delta = 6$ meV, similarly to CNTs[5,28], but with the spectral weight of twilight states $m = 0.68 \pm 0.04 \approx 4$ times larger[5] than that of the CNT dark states (Fig. 3e). This means that, while multiple excitonic states are present, they do not quench PL nearly as much as in CNTs.

The resulting emission is thus so bright that it reveals a series of different spectral features (Fig. 2d). All features of the optical response are reproduced by the sum of the ZPL, G and RBLM phonon sidebands, the second interband transition $E_{12}$[14], plus a low-energy excimer PL peak at 1865 meV (identified from bundled 4-CGNRs) and a wide Lorentzian contribution for the high-energy absorbance, respectively (Supplementary Fig. 7). The maxima of both PL and absorbance are separated by one G-band phonon from the ZPL and the periodicity of the narrow PL features, when fitted with a multiple peak fit with independent free parameters, matches the predicted RBLM phonons precisely (inset Fig. 2d). The RBLM and G phonon contributions deviate clearly from a purely Franck–Condon behaviour which produces PL and absorbance spectra as mirror images of each other (see Supplementary Text 1, Supplementary Table 2 and Supplementary Fig. 7 for Franck–Condon modelling results).

The optical spectra were analysed by considering the Debye–Waller factor $\alpha = \frac{J}{J+\Phi}$, which determines the probability of zero-phonon transitions[29] by comparing the integrated intensities of the ZPL $J(\omega)$ and of the phonon wing $\Phi(\omega)$. The type of electron-phonon coupling present is then revealed by the $T$-dependence of $\alpha$ (Fig. 3b). In purely Franck–Condon (FC) coupled systems, $\alpha$ is strongly $T$-dependent, while for Herzberg–Teller (HT) coupling $\alpha$ is $T$-independent[30]. In the presence of both FC and HT couplings, we expect $\alpha$ to be different for absorption and PL bands[31]. Our measurements at 77 K give $\alpha = 0.010 \pm 0.003$ for absorbance vs. $\alpha = 0.06 \pm 0.01$ for PL, showing that both types of coupling are actually

present, in agreement with theoretical predictions of GNRs with edge modulations (Fig. 1a)[14].

This analysis allows us to build a full energy level diagram, where the twilight states are separated from the usual bright ones, and a series of different vibronically coupled modes produce strong mixing with the electronic states and enhance the PLQE (Fig. 2d, 4).

### PLE and time-resolved PL spectroscopy

Photoluminescence excitation (PLE) spectra complete the understanding of the fundamental electron-phonon interaction for twilight states (Fig. 5a).

The three absorbance peaks observed confirm the assignment, but normalising the PLE spectra to the peak excitation energy shows abrupt variations of the brightest excitation: upon crossing the ZPL, the peak excitation energy shifts from $E_{exc} = 2.30$ eV (ZPL+1G) to 2.34 eV (ZPL+1G+1RBLM) (Fig. 5b). This observation unveils the mechanism producing emission: below the ZPL, creation of one G phonon is the dominant mechanism in both absorption and consecutive PL emission, whereas emission of an RBLM plus a G phonon dominates the absorption for ZPL emission. Close to the ZPL, the strong mixing of the G and RBLM modes also yields $E_{21} + G$ peak oscillations with an energy corresponding to the RBLM. The mechanism is not universal and is different for the second interband transition at $E_{exc} = 2.60$ eV, which is much stronger in the PLE compared to the absorbance data (Supplementary Fig. 8). Carriers excited into the second conduction band are more likely to relax internally from the second to the first conduction band at low $k$ values than to follow a multiphonon-assisted transition.

Finally, we reveal how the twilight states respond dynamically: In the first few nanoseconds, PL spectra show low-energy emission from vibronic coupling to the RBLM and G modes, while by 10 ns emission is dominated by the long-lived, higher-energy D and ZPL transitions directly to the ground state (Fig. 5c, d). This new strong emission process arises as carriers cross over to the twilight states mediated by vibronic coupling and emit at the higher energy zero-phonon line. The state lifetimes can be extracted from biexponential fitting of the time dependence of the signal (Fig. 5e), yielding $0.6 \pm 0.1$ ns for the bright states, and $3.6 \pm 0.3$ ns and $2.9 \pm 0.3$ ns for the ZPL and D-mode emission from the twilight ones (Fig. 4).

Overall, these results reveal GNRs as the carbon nanostructures that can overcome excitonic K-K′ valley quenching to enable bright optical emission. The 8% quantum yields achieved in these first experiments here are already beyond those of most nanocarbon materials[7,32–34]. More importantly, we reveal mechanisms that can be used to turn dark states into much brighter twilight ones, where the fundamental symmetry is altered by the perfect edge modulation obtained by molecular design. These twilight states not only yield high PL efficiencies, but also a superior resolution of the spectral features, so that access to the fundamental optical processes is available. It is notable that this level of detail is very unusual in extended carbon nanostructures and is achieved both in solutions and thin films, i.e. in conditions suitable for applications. These results open new experimental possibilities: topological and quantum processes can now be investigated not just by surface probe techniques[13], but also optically and with the associated time-resolved techniques[35,36]. Direct insight into the mechanisms, with internal relaxation processes dominated by phonons, already indicates long exciton lifetimes and associated exciton motion, offering a way to chemically tune the optical response. The chemical versatility of GNRs also opens up their use as a new family that can be integrated with established systems such as poly-acenes[37]. The GNR side groups can be modified to bind to biological structures, or bear quantum coherent units, making these systems an exceptionally versatile optical platform for bio-imaging,[3] optoelectronics[17,35,36] and quantum technologies[38,39].

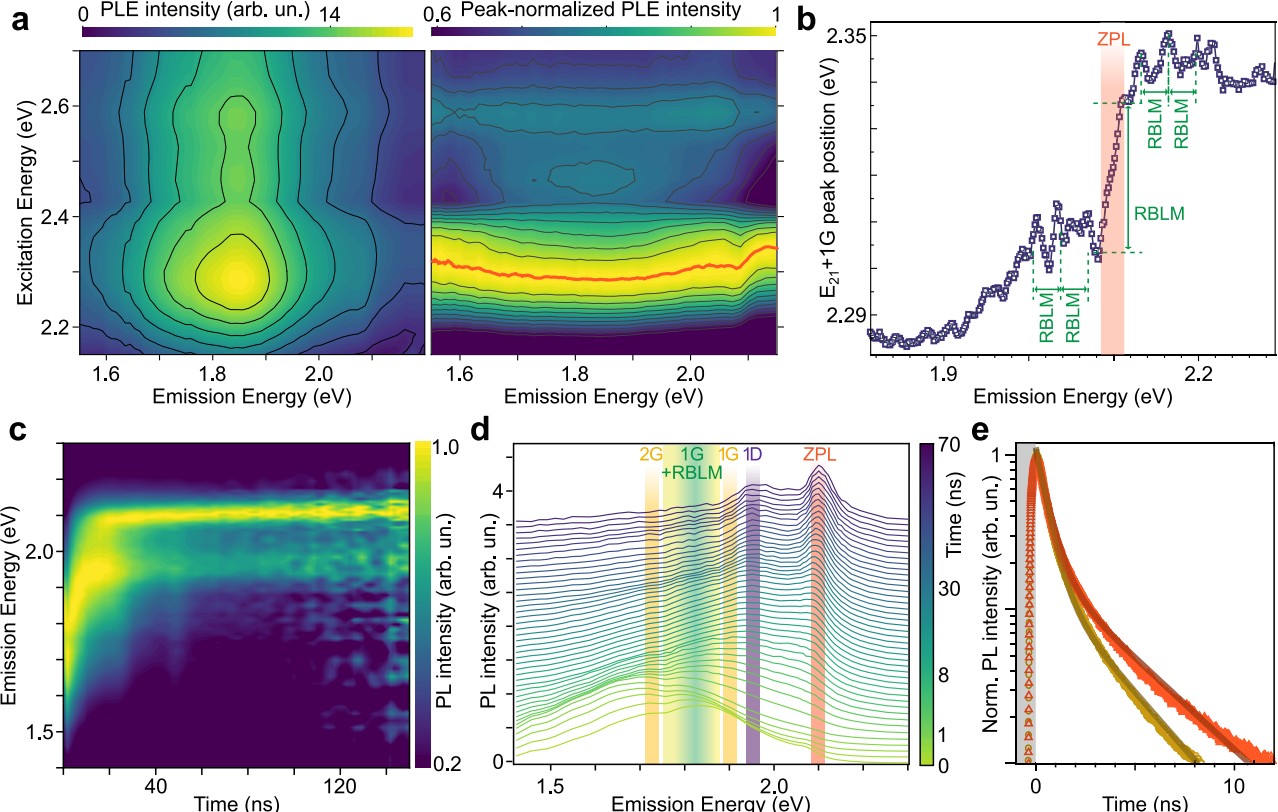

**Fig. 5 | Excitation energy and time dependence of GNR PL. a** PLE map of GNRs in chloroform solution, excited with a white light source (left) and its normalisation by peak intensity along the excitation energy axis (right). **b** Position of the $E_{21} + G$ peak for different emission energies, with the ZPL position (red) and the energy spacing of the RBLM (green). **c** Time evolution of the normalised PL spectra, acquired with 3.06 eV excitation in solution. **d** Temporal evolution of GNR modes. **e** Logarithmic plot of the 1G and ZPL PL intensities (yellow and orange, respectively) vs time, and fits to bi-exponentials (lines).

## Methods

Details of the sample preparation are provided in the Supplementary Methods. **Raman** spectra were acquired with a Jobin Yvon T64000 triple spectrometer and an Andor DU420A-OE CCD and excited with a Ventus solo Nd:YAG laser with excitation energy $E_{exc} = 2.33$ eV. **Absorption, photoluminescence and PLE** of GNRs were investigated in chloroform solutions (0.2 and 0.05 mg/ml, respectively) for $T = 295$ K and in transparent EVA polymer matrices at lower $T$. Absorbance measurements were conducted with a Perkin-Elmer UV/Vis-NIR spectrophotometer Lambda 35. PL data were acquired with a Princeton Applied Research Model 1235 triple grating 0.3 m spectrograph coupled to an Andor iDus 416 CCD and excited with a LaserQuantum Ventus532 532 nm (2.33 eV) laser or a 405 nm (3.06 eV) laser diode. PLE maps were measured at 295 K with a 75 W Xenon lamp (Photon Technology International Inc.) scanned by a monochromator and connected to a spectrometer leading to an InGaS photodiode array (OMA V, Princeton Instruments). Note that the Xenon lamp does not have sufficient power to operate at resolutions required to resolve the ZPL. **Time-resolved PL** was measured on a PicoQuant FluoTime 300 (4 ps resolution) and a PicoQuant LHD-P-C-405 (3.06 eV) pulsed laser diode at 4 MHz repetition rate on GNR in chloroform solution (0.05 mg/ml). **PLQE**. Photoluminescence quantum efficiency was measured with a Roithner MLL-III-405-200mW (3.06 eV) laser, an integrating sphere and a calibrated QEPro OceanInsight grating spectrometer on GNR in chloroform solution.

### First principle calculations

Density Functional Theory (DFT) calculations were implemented in SIESTA[40]. We employed Perdew-Burke-Ernzerhof (PBE) generalized gradient approximation (GGA) functional[41] in the calculation of the

band structure. Energy cut-off of 400 Ry and the Monkhorst-Pack grid of (21,1,1) were used. A vacuum region of at least 18 Å is added in nonperiodic directions to prevent unwanted interactions. We removed the alkyl chain in the side groups to simplify the calculation. The structure was optimized until the maximum force on the atoms is less than 0.01 eV/Å. In the Density of States (DOS) calculation gaussian broadening of 0.05 eV is used for all bands.

In the phonon calculation, we adopted tighter criteria for structure relaxation and convergence. Perdew-Zunger (PZ) local density approximation (LDA) functional[42] was used. The energy cut-off was increased to 800 Ry. Monkhorst-Pack k-sampling was set to (50, 1, 1). Due to the computational complexity, we further simplify the result and only keep anthracene in the side groups. The structure was relaxed with maximum allowed force of 0.004 eV/Å Then a supercell of [3, 1, 1] was created for calculating phonon dispersion.

## Data availability

The data supporting the findings of this study are available online within the Oxford University Research Archive at the Bodleian Library of Oxford[43]. No custom code is used.

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

## Acknowledgements

We acknowledge funding from EPSRC (CDT-PV-EP/L01551X/1), University College, Oxford (Oxford-Radcliffe scholarship), EU (ERC-StG 338258 OptoQMol, ERC-CoG 819698 T2DCP, ERC-CoG MMGNRs, Marie-Curie-IF 894761 MolecularMAGNET, Marie-Curie-ITN 813036 ULTIMATE, FET 101017821 LIGHT-CAP, Pathfinder-101099676-4D-NMR, and GrapheneCore3 881603), the Royal Society (University Research Fellow and URF grant), the Max Planck Society and machine time by the Oxford Advanced Research Computing facility.

## Author contributions

BS performed the optical measurements. FK performed the numerical calculations. XY and WN prepared the samples supervised by JM, LB and XF. RN and LB conceived the experiments. BS, FK, RN and LB analysed the data, developed the interpretative framework, and prepared the

manuscript. LB and BS made the figures. RN, MR and LB supervised the work. All authors discussed the final manuscript and provided feedback.

## Competing interests

The authors declare no competing interests.
