## [Peer Review File · Nature Communications]

Emissive brightening in molecular graphene nanoribbons by twilight statesEditorial Note: This manuscript has been previously reviewed at another journal. This document only contains reviewer comments and rebuttal letters for versions considered at *Nature Communications*.

REVIEWER COMMENTS

Reviewer #1 (Remarks to the Author):

The revised manuscript from Sturdza, Bogani, and Nicholas et al. has been significantly improved. Most of the concerns raised by reviewers have been properly addressed. In my opinion, this photophysics-oriented work has provided valuable insights for the general audience to achieve enhanced quantum yields of carbon materials.

One suggestion: the authors seem to misuse the term "polydispersity", which in the chemistry community exclusively means the length distribution of polymers. In the SI, "Ribbons have a length distribution of $L=10-371$ nm." In this regard, the polydispersity of the studied materials is really large and there is no approach adopted in the current manuscript to "reduce" the polydispersity. Therefore, the authors cannot write that "we study synthetically-made graphene nanoribbons with reduced polydispersity". The "structural polydispersity" mentioned by authors, in my understanding, may refer to the precise chemical structure of the studied GNRs with the same width and edge. The discussions in the main text should also be revised.

Reviewer #2 (Remarks to the Author):

The authors have appropriately addressed the comments I had raised in the previous review round. Authors have also satisfactorily responded to the comments of other reviewers. Hence, I recommend the work in its current form to be published in *Nature Communications*.

Reviewer #3 (Remarks to the Author):

The authors have provided an answer to my comment and have modified their manuscript accordingly. As I wrote in my original letter, I found the paper nicely written of major importance with its main claims well supported. I therefore support its publication in *Nature Communication*. Among the points I rose, I am however still not satisfied by the fitting procedure proposed in Figure 2d (and FigS7) and the associate claim that the spectra reveal "well-defined features" that they are of "high resolution", that they reveal a "plethora of spectral features", and, on a more quantitative aspect, "that the resolution is certainly good enough to assign multiple peaks" (in the authors response letter).

Let me detail my point: The ZPL feature is well resolved, much better than in any other paper published so far reporting on PL measurement on GNRs. This is extremely nice and as long as the comments of the authors discuss the resolution of this specific peak I have no problem. This is actually one of the reason why I think this paper deserves publication in *Nature Com*.

However, I am still not convinced by the fitting procedure used to identify the RBML modes in the 4K PL spectrum of Figure 2d. Excluding the ZPL, the low energy part of the spectrum (where the RBML modes supposedly lie) is fitted with no less than 7 peaks, as detailed in Fig S7. One peak is attributed to the G mode, five to the RBML, and one (at lower energy) to an excimer emission (if I understand correctly). This last one already is absent from the Fig2d and should definitively be added.

More importantly, with 8 Lorentzian peaks one can fit nearly anything when the features in the experimental spectrum are barely distinguishable. I am convinced that other linear combination of

Lorentzian peaks might give an overall quite accurate fit.

Going deeper into this discussion, I wonder why the authors consider different widths for their five RBLM modes? As they all belong to the same vibronic progression, I would find logic to fit the 5 RBLM peaks with the same width (which ideally should be the same than the ZPL). What about the respective intensities of the 5 RBLM peaks, should they not follow some given pattern (being due to a FC progression or to some other rules)?

I would not be so critical if the authors would present their fits in Fig S7 fit with great care, explaining the limit of such fitting procedure and then use it to suggest that these features might be related to a progression of the RBML modes. But this is not how it is presented now.

Smaller (but related) point, I do not understand why two different "strategies" are used to fit the 4K and 77K data (still in Fig S7). The authors should probably clarify that.

REVIEWER COMMENTS

Reviewer #1 (Remarks to the Author):

The revised manuscript from Sturdza, Bogani, and Nicholas et al. has been significantly improved. Most of the concerns raised by reviewers have been properly addressed. In my opinion, this photophysics-oriented work has provided valuable insights for the general audience to achieve enhanced quantum yields of carbon materials.

One suggestion: the authors seem to misuse the term "polydispersity", which in the chemistry community exclusively means the length distribution of polymers. In the SI, "Ribbons have a length distribution of $L=10-371$ nm." In this regard, the polydispersity of the studied materials is really large and there is no approach adopted in the current manuscript to "reduce" the polydispersity. Therefore, the authors cannot write that "we study synthetically-made graphene nanoribbons with reduced polydispersity". The "structural polydispersity" mentioned by authors, in my understanding, may refer to the precise chemical structure of the studied GNRs with the same width and edge. The discussions in the main text should also be revised.

We thank the reviewer for pointing this out. We agree that the term 'polydispersity' might confuse some readers and have thus addressed this issue in the manuscript. We have replaced it in lines 150ff by the following:

'This observation is exceptional for ensemble measurements on carbon nanostructures, where the ZPL is usually broadened by polydispersity, disorder in the individual nanostructures such as edge and width fluctuations, and inter-structure energy transfers. The optical response therefore demonstrates the superb level of cleanliness of the de-bundled GNRs.'

We replaced 'polydispersity' by 'edge and width disorder' in line 91.

Reviewer #2 (Remarks to the Author):

The authors have appropriately addressed the comments I had raised in the previous review round. Authors have also satisfactorily responded to the comments of other reviewers. Hence, I recommend the work in its current form to be published in Nature Communications.

We thank reviewer 2 for their feedback.

Reviewer #3 (Remarks to the Author):

The authors have provided an answer to my comment and have modified their manuscript accordingly. As I wrote in my original letter, I found the paper nicely written of major importance with its main claims well supported. I therefore support its publication in Nature Communication. Among the points I rose, I am however still not satisfied by the fitting procedure proposed in Figure 2d (and FigS7) and the associate claim that the spectra reveal "well-defined features" that they are of "high resolution", that they reveal a "plethora of spectral features", and, on a more quantitative aspect, "that the resolution is certainly good enough to assign multiple peaks" (in the authors response letter).

We have replaced 'plethora' with 'series'.

Let me detail my point: The ZPL feature is well resolved, much better than in any other paper published so far reporting on PL measurement on GNRs. This is extremely nice and as long as the comments of the authors discuss the resolution of this specific peak I have no problem. This is actually one of the reason why I think this paper deserves publication in Nature Com.

However, I am still not convinced by the fitting procedure used to identify the RBLM modes in the 4K PL spectrum of Figure 2d. Excluding the ZPL, the low energy part of the spectrum (where the RBLM modes supposedly lie) is fitted with no less than 7 peaks, as detailed in Fig S7.

One peak is attributed to the G mode, five to the RBLM, and one (at lower energy) to an excimer emission (if I understand correctly)

This is correct.

This last one already is absent from the Fig2d and should definitively be added.

Agreed, we have added the Excimer peak to Fig 2d.

More importantly, with 8 Lorentzian peaks one can fit nearly anything when the features in the experimental spectrum are barely distinguishable. I am convinced that other linear combination of Lorentzian peaks might give an overall quite accurate fit.

Going deeper into this discussion, I wonder why the authors consider different widths for their five RBLM modes? As they all belong to the same vibronic progression, I would find logic to fit the 5 RBLM peaks with the same width (which ideally should be the same than the ZPL). What about the respective intensities of the 5 RBLM peaks, should they not follow some given pattern (being due to a FC progression or to some other rules)?

I would not be so critical if the authors would present their fits in Fig S7 fit with great care, explaining the limit of such fitting procedure and then use it to suggest that these features might be related to a progression of the RBLM modes. But this is not how it is presented now. Smaller (but related) point, I do not understand why two different “strategies” are used to fit the 4K and 77K data (still in Fig S7). The authors should probably clarify that.

We thank the reviewer for this opportunity to clarify the fitting procedure, we have included the text below in the updated version of Supplementary Text S1.

To fit the optical response, we chose a staged process:

First, we performed a free parameter fit on amplitude, width, and position in order to determine the energy of the transitions and hence the basic mechanism (top of Fig. S7). For the PL, this requires eight Lorentzian peaks, one for the ZPL, five for the RBLM-mode, one for the G-mode, and one for low energy excimer emission. For the absorption, we chose nine Lorentzian peaks, one for the ZPL, three for the RBLM-mode, three for the G-mode, one for the E_{12} transition, and one for high energy absorption.

Fitting a single spectrum with this many parameters leaves room for uncertainty in the fitting result as this often leads to multiple solutions with similar accuracy. However, we found that when all parameters are free and independent, the results shown in Fig. S7 were robustly reproducible under multiple varying starting parameters.

This procedure shows us that the ZPL and 5 RBLM peaks form a single series of peaks with a very accurate spacing equal to the RBLM energy as shown in the inset of Fig. 2d.

In the second step, we chose a fit function satisfying a simple Franck-Condon model which imposes conditions on the position, width and amplitude of the phonon peaks. For the absorption spectrum, which is dominated by the G-mode, this works relatively well.

However, the PL spectrum deviates from the predicted amplitudes of a purely Franck-Condon model and we have thus included an extra peak for the 0-1 RBLM phonon (Fig. S7) to allow the fit function to match the data sufficiently well. Still, the fit function deviates from the data at energies approaching the G-mode peak around 1900meV. The higher RBLM modes at this point are significantly enhanced relative to the simple Franck-Condon model which suggests the presence of a second phonon coupling mechanism such as Herzberg-Teller coupling, as discussed in the main text.

REVIEWERS' COMMENTS

Reviewer #3 (Remarks to the Author):

The authors have accounted for my comments and questions. The paper can now be published without further modifications.